# Plant Reproductive Success Mediated by Nectar Offered to Pollinators and Defensive Ants in Terrestrial Bromeliaceae

**DOI:** 10.3390/plants13040493

**Published:** 2024-02-08

**Authors:** Carolina Torres, Mariana P. Mazzei, José L. Vesprini, Leonardo Galetto

**Affiliations:** 1Instituto Multidisciplinario de Biología Vegetal (IMBIV), Consejo Nacional de Investigaciones Científicas y Técnicas, Universidad Nacional de Córdoba, Vélez Sarsfield 1611, Córdoba CP X5016GCA, Argentina; ctorrres111@yahoo.com.ar; 2Cátedra de Diversidad Biológica III, Facultad de Ciencias Exactas, Físicas y Naturales, Universidad Nacional de Córdoba, Vélez Sarsfield 299, Córdoba CP 5000, Argentina; 3Instituto de Investigaciones en Ciencias Agrarias de Rosario, IICAR-CONICET-UNR, CC 14, Zavalla CP S2125ZAA, Santa Fe, Argentina; mariana.mazzei@unr.edu.ar (M.P.M.); jvesprin@unr.edu.ar (J.L.V.)

**Keywords:** ant-excluded plants, compatibility system, *Dyckia*, floral nectar, extrafloral nectar, fruit set, seed set, animal–plant mutualisms

## Abstract

Most plants produce floral nectar to attract pollinators that impact pollination and seed production; some of them also secrete extrafloral nectar harvested by insects that may influence the plant reproductive success. The aim of this study was to analyze the effects of excluding pollinators and/or ants on the per-plant reproductive success in two species (*Dyckia floribunda* Griseb. and *Dyckia longipetala* Baker, Bromeliaceae) that produce floral and extrafloral nectar. The hypothesis states that both ecological processes (pollination and ant defense) involving nectar-mediated animal–plant interactions are beneficial for plant reproductive success. We expected the highest decrease in the plant fruit and seed sets when the pollinators and ants were excluded, and a moderate decrease when solely ants were excluded, compared to the control plants (those exposed to pollinators and ants). In addition, a lower natural reproductive success was also expected in the self-incompatible *D. longipetala* than in the self-compatible *D. floribunda*, as the former totally depends on animal pollination for seed production. *D. floribunda* and *D. longipetala* presented similar trends in the response variables, and the expected results for the experimental treatments were observed, with some variations between species and among populations. The ecological function of nectar is important because these two plant species depend on pollinators to produce seeds and on ants to defend flowers from the endophytic larvae of Lepidoptera. The study of multispecies interactions through mechanistic experiments could be necessary to clarify the specific effects of different animals on plant reproductive success.

## 1. Introduction

Nectar (hereafter floral and extrafloral to differentiate nectar linked to pollinators or other visitors, respectively) is presented to animals during the reproductive cycle of a plant to mediate different ecological processes. Nectar is an aqueous sugar solution secreted by the floral nectaries of most flowering plant species [1,2] that mediates pollination. Some plants also secrete nectar through extrafloral structures, mediating other ecological processes, such as defenses against herbivores [3,4,5]. While plant–pollinator interactions are positive (i.e., mutualism) for both sides, ant–plant interactions may vary from positive to negative or even show no protective effects against herbivores [6,7,8]. For example, plant reproductive success can be modified by variations in ant–plant interactions though time and/or space [4,7], by the aggressiveness of some ant species to some pollinators, reducing the frequency of bee visits [9], or by decreasing the seed production [10]. Thus, it is important to analyze the differential contributions of pollinators and ants to the per-plant reproductive success.

Species of Bromeliaceae secrete floral nectar through septal nectaries [11], and they are pollinated by a wide spectrum of animals [11,12,13,14]. Some Bromeliaceae species may present undifferentiated extrafloral nectaries in the floral perianth, which secrete nectar through stomata with a high concentration, which attracts ants [15,16]. We selected two Bromeliaceae species, *Dyckia floribunda* Griseb. and *Dyckia longipetala* Baker, to study the effects of multispecies animal–plant interactions mediated by nectar on the plant reproductive success. These two *Dyckia* species present non-structured nectaries that secrete extrafloral nectar on the calix that attracts other visitors [15]. Ants crop the extrafloral nectar of both species as soon as it is secreted, and the secretion process seemed to be continuous because the droplets keep growing in diameter if they are not removed (Figure 1A,B,E). Ants visit extrafloral nectaries during both the flowering and fruiting periods ([15]; Figure 1C,D). In a previous study, we reported that ants are attracted to extrafloral nectaries and can protect the reproductive organs of *D. floribunda*, as the ant-excluded plants produced a lower number of fruits compared to the control plants [17]. The floral nectary produces a mean nectar volume of 4–6 µL per flower (a 31–33% solute concentration and 1.2–1.6 mg of sugar per flower). Floral nectar is secreted continuously during the anthesis period in these two species, and it is not reabsorbed in the unvisited flowers [18]. In the sugar composition of the floral nectar in these species, sucrose (46–49%) predominate over hexoses [12]. The sugar composition of extrafloral nectar is different from that of floral nectar because it is composed of almost pure sucrose (from 99.5 to 100%; [15]). Preliminary observations showed that *D. floribunda* could be self-compatible and *D. longipetala* could be self-incompatible.

Our hypothesis states that both ecological processes involving nectar-mediated animal–plant interactions (pollination and plant protection by ants) are beneficial for the plant reproductive success. The effects of the animal–plant interactions on the fruit and seed sets were evaluated with the following experimental treatments: bagged plants (pollinators and ants were excluded); ant-excluded plants (pollinators could access the flowers); and control plants (naturally pollinated and visited by ants). We expected the highest decrease in the per-plant fruit and seed sets when the pollinators and ants were excluded, and a moderate decrease in the plants when solely ants were excluded, compared to the control plants (those exposed to pollinator and ant visits). In addition, if the preliminary observations of the compatibility system were confirmed for these two species, we expected a lower natural reproductive success in the self-incompatible *D. longipetala* than in the self-compatible *D. floribunda*, as the former has a higher dependence on animal pollination for fruit and seed production, independently of the role of ants in plant protection.

## 2. Results

### 2.1. Ant Presence and Behavior

Nine morphospecies of ants were registered on individuals of both plant species collecting extrafloral nectar on flowers or on developing fruits during the reproductive cycle. *D. longipetala* and *D. floribunda* showed means of 2.25 and 3.78 ants per spike, respectively (n = 32 and 52 plants monitored during the reproductive season, respectively; ranges of 0–10 and 0–23 ants per spike, respectively). The most common ants were *Camponotus rufipes* (Figure 1C) and *Crematogaster quadriformis* (Figure 1D). Many individual ants were usually seen at the same time on the spikes, but they were usually of only one species. Large ants deterred insect visitors, particularly *C. rufipes*, including some of the floral visitors (e.g., small bees or diurnal butterflies), when they visited the flowers. The flowers and fruits can be predated by two main groups of endophytic larvae that ants cannot eliminate. Lepidoptera (Noctuideae) larvae were observed consuming seeds inside of the developing fruits of both plant species. Coleoptera (Curculionidae) larvae were found inside the flower buds of *D. floribunda* consuming the ovaries and stamens before flower opening. 

### 2.2. Floral Pollinators and Nectar Standing Crop

The flowers of *D. longipetala* were frequently visited by hummingbirds (*Chlorostilbon lucidus* and *Heliomaster furcifer*; 0.10 and 0.17 visits per 15 min period, respectively). Usually, these pollinators visited a few open flowers per plant for short periods (< 10 s per plant visit). *D. floribunda* showed a more diverse assemblage of floral visitors than *D. longipetala*. *Chlorostilbon lucidus* (Appendix A), *Sappho sparganura*, and *Bombus* spp. were the most frequent visitors of *D. floribunda* (0.55, 0.12, and 0.32 visits per 15 min period, respectively). These pollinators probed a wide range of flowers during a visit; for example, they visited a few open flowers per plant (<5 s per plant visit) or most open flowers of the spike (ca. 60 s per plant visit). Hummingbirds (mostly *Ch. lucidus*) usually displayed territorial behavior, defending a patch of many flowering plants of *D. floribunda*. *Heliomaster furcifer* occasionally visited this species, as well as *Heraclides thoas*, *Danaus erippus*, *Vanessa* sp. (Appendix A), and another morphospecies of diurnal and nocturnal (Appendix A) lepidopterans. *Apis mellifera* and Halictideae species collect pollen from *D. floribunda* and can pollinate flowers because the anthers and stigma are in proximity (Appendix A).

In general, the floral nectar standing crop was low, and most flowers had almost no nectar due to the high frequency of pollinator visits, except during the early morning hours. The floral nectar standing crop in *D. floribunda* ranged between 0 and 5.4 µL per flower (n = 30 plants), with a nectar concentration ranging between 17.1 and 52.3%. *Dyckia longipetala* showed a comparable nectar volume per flower (the standing crop ranged from 0 to 6.3 µL per flower; n = 14 plants) but with a lower nectar concentration (ranging from 15.5 to 37.2%).

### 2.3. Plant Reproductive Compatibility System

The hand pollinations showed that *D. longipetala* is self-incompatible. Hand self-pollinations (n = 37 flowers) were performed on nine different plants and no fruits developed (0%). The hand cross-pollinations showed a higher reproductive success (28 fruits were developed from 30 cross-pollinated flowers: 93%). Self-tubes were observed within the style, but the pollen tube growth was interrupted ca. 8 mm below the stigmatic tissue.

*Dyckia floribunda* could produce fruits after the self-pollinations (137 fruits/470 hand-self-pollinated flowers: 29%; n = 17 plants). The hand cross-pollinations showed a higher reproductive success rate (86%: a total of 42 developed fruits from 49 pollinated flowers). Observations of the self-pollen tubes in this species indicated that their growing rate is comparable to that of cross-pollen tubes. 

### 2.4. Experimental Design Linking Pollination, Defensive Ants, and Plant Reproductive Success

*Dyckia longipetala* consistently achieved lower values than *D. floribunda* for the ‘fruit set’ and ‘seeds per plant’ in all treatments (Table 1; Figure 2 and Figure 3). When pollinators and ants visited the spikes of both species (‘exposed’ treatment), the highest values for the ‘fruit set’ and ‘seeds per plant’ were recorded (Figure 2 and Figure 3; Table 1 and Table 2). The ‘exposed’ treatment exhibited greater variation in the fruits and seeds between the populations of both plant species than the other treatments (Figure 2 and Figure 3). When ants were excluded, the ‘fruit set’ and ‘seeds per plant’ showed lower values than the ‘exposed’ plants (*p* < 0.001 for both species) but higher values than the ‘bagged’ plants (significant differences were recorded for only one of these two species; *p* > 0.05 and *p* < 0.01 for *D. longipetala* and *D. floribunda*, respectively; Figure 2 and Figure 3; Table 1 and Table 2). The natural (exposed plants) fruit set and total number of seeds per plant were lower in the self-incompatible species, *D. longipetala*, compared to the self-compatible species, *D. floribunda* (Table 1; Figure 2 and Figure 3).

## 3. Material and Methods

### 3.1. Plant Material

We selected two plants for this study: *Dyckia floribunda* and *Dyckia longipetala*; these species are distributed in the Argentinian Chaco ([11]; Appendix A) and produce flowers once a year. 

### 3.2. Plant Reproductive Compatibility System

We studied the compatibility system to better understand the experimental results of the treatments, comparing the effects of the ants and pollinators on the reproductive success of the plants. Compatibility tests were performed to confirm whether the individuals of these two species in the studied populations presented the expected pattern according to the preliminary observations. We used two complementary approaches to study the compatibility system: (a) hand-controlled pollinations followed until fruit development and (b) hand-controlled pollinations to analyze the pollen-tube growth. The flowers of inflorescences from 17 and 9 plants for *Dyckia floribunda* and *D. longipetala*, respectively (see details for the number of flowers for each pollination treatment in the Section 2), were as follows: (i) flowers self-pollinated with their own pollen and (ii) flowers outcross-pollinated with the pollen of plants located at a > 20 m distance. Self- and cross-pollinations were performed on 2–4 tagged flowers in each inflorescence within the first five hours after flower opening. Five flowers of each treatment (one per treatment of 5 different plants) were removed from the plants after hand pollinations (6–18 h) and gynoecia were placed and stored in 70% ethanol (porta, Córdoba, Argentina). Squashes were performed according to [19], with some modifications: gynoecia were softened with sodium hydroxide (0.8 N; Sigma-Aldrich, Saint Louis, MO, USA) for 3–4 h, washed twice with distilled water, stained with aniline blue (Sigma-Aldrich, Saint Louis, MO, USA) for 15–30 min, and squashed under a coverslip to spread the stylar tissue. Pollen tubes were observed under an epifluorescence microscope (Zeiss Axiophot equipped with UV filters; Carl Zeiss, Göttingen, Germany) measuring the distance that the pollen tubes had grown into the style with a digital caliper. The remaining hand-pollinated flowers were tagged and followed until complete fruit maturation.

### 3.3. Experimental Design

Experiments were conducted in the Chaco Serrano (Argentina, Córdoba, Depts. of Colón and Punilla). Three populations (El Diquecito, Río Ceballos, and Cosquín) were studied from October to December, from flowering to fruiting, before seed dispersal.

From 15 to 20 plants per population of each species were selected to perform observations and experiments. Plants were tagged and randomly assigned to three experimental treatments when the inflorescence began to develop. Individuals 1 m apart were used to avoid choosing plants from the same genet. 

The treatments consisted of the following:Exposed inflorescences (control treatment): ants and pollinators had free access;Ant-excluded inflorescences: Spikes prevented ant access and pollinators had free access. Before flowering, the base of the spike was coated with an oil-based insect repellent (Tanglefoot^®^, Grand Rapids, MI, USA). The repellent was kept at least 20 cm away from the nearest flower, a reasonable distance to avoid its influence on pollinators and flying herbivores. To avoid the formation of natural bridges for ants to the upper sections of the spikes on the ant-excluded plants, the adjacent stems of neighboring plants were removed;Bagged inflorescences: ants and pollinators were excluded from the spikes using a voile.

Treated (ant-excluded and bagged) plants were checked weekly until fruit maturation for the repellent reposition or for the bag integrity. Spikes from all treatments were collected before seed dispersal. In the laboratory, the total number of fully developed fruits produced by each spike was counted. The seeds were counted in a sample of 7–15 fruits from each infructescence. Finally, the total number of seeds per infructescence was estimated by multiplying the total number of developed fruits by the mean number of seeds per fruit.

### 3.4. Pollinators, Ants, and Insects Consuming Plant Tissues

We recorded the frequency and identity of the pollinators (numbers of visits per 15 min periods along the flowering season for exposed plants; totals of 17 and 11 h of observations for *Dyckia floribunda* and *D. longipetala*, respectively), the number and species (or morphospecies) of ants on the spikes, and the floral nectar standing crop as an indirect indicator of pollinator visits. Nectar standing crop was measured using graduated capillaries (Drummond Scientific Co., Broomall, PA, USA) and a hand-refractometer (Atago Co., Tokyo, Japan). We occasionally performed observations of animals in the field consuming plant reproductive organs; the vegetative organs of these species are not consumed because the leaves are spiny and present a very thick cuticle. 

### 3.5. Data Analysis: Linking Pollination, Defensive Ants, Plant Reproductive Success, and Compatibility System

We employed Generalized Linear Mixed Models to examine the relationship between the independent [‘treatments’ (‘levels: ‘bagged’, ‘ant-excluded’, and ‘exposed’) and ‘species’ (levels: *‘D. floribunda*’ and ‘*D. longipetala*’)] and the dependent (‘fruit set’ and ‘seeds per plant’) variables. The models followed a Gaussian distribution, with the nested random effects estimating the intercepts of all ‘plant’ levels measured within each ‘population’ level. Statistical analysis was performed using R statistical software, version 4.2.1 [20], in combination with specific packages. The model was fitted using the ‘lme4’ package with restricted maximum likelihood estimation [21]. Pairwise comparisons between treatments and species were performed using the ‘emmeans’ package, and *p*-values were adjusted using the Tukey method [22]. The graphs were generated with the ‘ggeffects’ and ‘ggplot2’ packages [23,24]. 

## 4. Discussion

The presence of pollinators and ants attracted by floral and extrafloral nectar has positive outcomes on the reproductive success of these two *Dyckia* species. The exclusion of pollinators and ants showed the highest effects for decreasing the fruit and seed sets, particularly in the self-incompatible *D. longipetala*. When solely ants were excluded, the reproductive output decreased in both species, but in lower percentages than in the bagged plants. 

Although the general trends between species are concurrent, some variations for the fruit and seed sets produced per plant were observed between the species and among the population of the same species. The floral nectar standing crop and frequency of visits indicate that pollinators guarantee pollination and seed production in both *Dyckia* species, although there are differences in the compatibility systems between them. The higher values for the fruit and seed sets observed in *D. floribunda* in comparison to *D. longipetala* could be related not only to the differences in their compatibility systems (self-compatible and self-incompatible, respectively) but also to other factors that were not measured in this study (e.g., negative interactions between ants and pollinators, the consistency of the ant–plant interactions in time and/or space, nectar costs, etc.). 

The consequences of ant activity on the plant reproductive success can differ and are classified as positive [3,25,26,27,28], neutral [8], and negative. In the latter case, the presence of ants may reduce the frequency of pollinator visits due to the aggressiveness of the ants or may decrease the pollen available for pollination [5,6,7,9,10,29,30]. Although our results showed the benefits of ants to the plant reproductive success, some of the ant species (*C. rufipes*) can be aggressive to some pollinators (mainly small bees and lepidopterans) in *D. floribunda* and may reduce the pollination when many individual ants are simultaneously on the spikes looking for extrafloral nectar. The location of extrafloral nectaries in Bromeliaceae (near the reproductive organs [15]) could be a trade-off because ants could deter some pollinators, and because plants with extrafloral nectaries in reproductive organs, in general, benefit more from ant attendance compared to plants bearing extrafloral nectaries on vegetative organs [31]. Nevertheless, these *Dyckia* species are pollinated mostly by hummingbirds, and ants do not seem to affect their floral visits. It is necessary to perform manipulative experiments with ants in these *Dyckia* species to clarify the effects of ants on pollinators and pollination. In a previous study with this approach, in which the presence of ants (*C. rufipes*) on the inflorescences of a mistletoe with hummingbird-pollinated flowers was manipulated to assess their effects on the behavior of this pollinator, no effects on the visitation rates of the pollinators were found [32]. 

The ecological function of extrafloral nectar for the individual plant seems to be important in these Bromeliaceae species because it is secreted from early bud development to full fruit maturation. This prolonged period of ant–plant interaction mediated by extrafloral nectar could be justified because some animals feed on the ovaries and anthers and others feed on the seeds. The variations in the plant reproductive success within the ant-excluded treatment (and between populations) may be explained because ant–plant interactions mediated by extrafloral nectar are not consistent in time and/or space [28,33,34]. For example, plants can be attended by different species of ants, or different numbers of ants of the same species can be observed on the spikes of different plants, or the ants can vary their number throughout the day on the same spike. Thus, ant–plant interactions are highly variable in time and space because ant species cannot prevent the presence of diverse animals feeding on different tissues (flowers, fruits, leaves, etc.) [34,35,36,37,38]. 

The costs of nectar production on the plant reproductive success could be high at the individual [39] or flower level [40], depending on the physiological strategy of nectar secretion of the species. The dynamics of floral nectar secretion and the response to experimental successive nectar removals may vary among Bromeliaceae species [13,18,41]. However, these two *Dyckia* species do not change their total floral nectar production after experimental removals and do not reabsorb nectar at the end of flower anthesis [18]. Extrafloral nectar can be secreted continuously, or secretion can be induced by the presence of herbivores to enhance the ant defense [42,43]. Extrafloral nectar appears to be secreted continuously in these two Bromeliaceae species, as droplets of almost pure sugars are accumulated and dehydrated on the floral perianth when ants are excluded from the spikes. The continuous nectar secretion pattern, for both floral and extrafloral nectar that attracts pollinators and defensive ants, suggests that the benefits for plant reproduction are greater than the nectar production costs. 

The results suggest that pollinators and ants have complementary positive effects on the reproductive success of both *Dyckia* species, and, accordingly, the study of multispecies interactions is necessary to better understand the process of seed production in plants. New studies comparing the variations in the reproductive success of many pairs of congeneric species with contrasting compatibility systems that are visited by ants and pollinators could strengthen this preliminary and interesting trend for these two *Dyckia* species. Moreover, the effects of the trade-offs between ants and pollinators on plant reproductive success need to be analyzed in more detail through a mechanistic approach to determine the causal explanations for the variations in the fruit and seed production between species and among populations.

## Figures and Tables

**Figure 1 plants-13-00493-f001:**
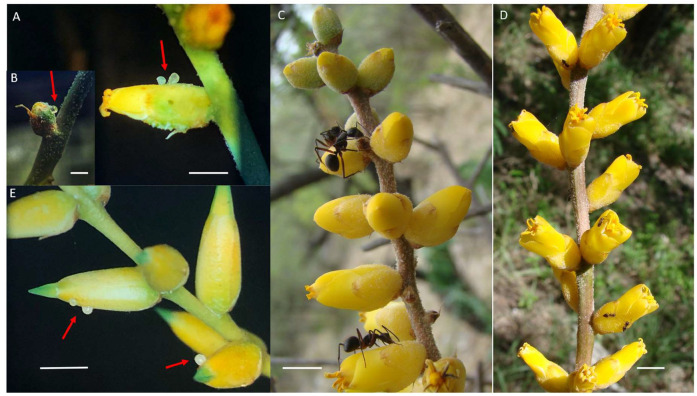
Extrafloral nectar in *Dyckia floribunda* and *D. longipetala*. (**A**–**D**) *D. floribunda*: (**A**) the arrow indicates two drops accumulated on the perianth of a flower in anthesis; (**B**) the arrow indicates a drop accumulated on the perianth of a bud at an early stage of development; (**C**,**D**) *Camponotus rufipes* and *Crematogaster quadriformis*, respectively, collecting extrafloral nectar. (**E**) The arrows indicate drops accumulated on nearly open flower buds of *D. longipetala*. Bars = 5 mm.

**Figure 2 plants-13-00493-f002:**
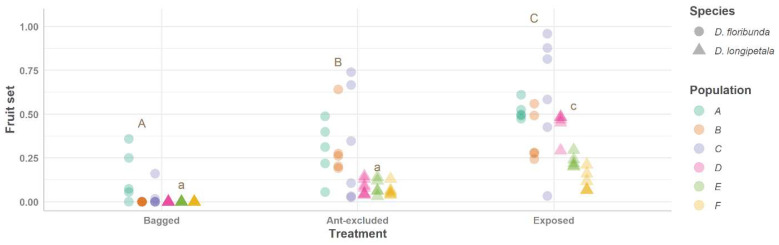
Fruit sets for *Dyckia floribunda* (dots) and *D. longipetala* (triangles) comparing experimental treatments (bagged plants, ant-excluded plants, and plants exposed to pollinators and ants). Values of fruit set for each treatment are presented showing all plants measured (each dot or triangle). Results of adjusted models are presented showing data for six different populations (random factor; see Table 1 for details): A–C for *D. floribunda* and D–F for *D. longipetala*, indicated with different colors. Different letters indicate differences between treatments for each species (capital letters for *D. floribunda* and lowercase letters for *D. longipetala*).

**Figure 3 plants-13-00493-f003:**
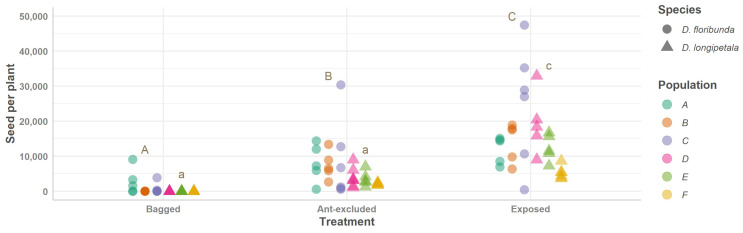
Numbers of seeds per plant for *Dyckia floribunda* (dots) and *D. longipetala* (triangles) comparing experimental treatments (bagged plants, ant-excluded plants, and plants exposed to pollinators and ants). Values of numbers of seeds per plant for each treatment are presented showing all plants measured (each dot or triangle). Results of adjusted models are presented showing data for six different populations (random factor; see Table 1 for details): A–C for *D. floribunda* and D–F for *D. longipetala*, indicated with different colors. Different letters indicate differences between treatments for each species (capital letters for *D. floribunda* and lowercase letters for *D. longipetala*).

**Table 1 plants-13-00493-t001:** Predicted values for ‘fruit set’ and ‘seeds per plant’ are provided per species of *Dyckia* and experimental treatment, as well as means for each treatment and their 95% confidence intervals.

Species	Variables	Treatments	Means	Confidence Intervals [95% for Predicted Values]
*Dyckia longipetala*	Fruit set	Bagged	0	[−0.04, 0.04]
		Ant-excluded	0.08	[0.04, 0.12]
		Exposed	0.25	[0.21, 0.30]
	Seeds per plant	Bagged	0	[−2315.05, 2337.28]
		Ant-excluded	3209.11	[1015.86, 5402.36]
		Exposed	11,851.02	[9535.97, 14,188.30]
*Dyckia floribunda*	Fruit set	Bagged	0.06	[−0.06, 0.17]
		Ant-excluded	0.31	[0.20, 0.42]
		Exposed	0.51	[0.40, 0.62]
	Seeds per plant	Bagged	1126.29	[−3131.42, 5558.08]
		Ant-excluded	8091.56	[3833.85, 12,523.35]
		Exposed	17,453.15	[13,195.44, 21,884.94]

**Table 2 plants-13-00493-t002:** Pairwise comparisons between ‘treatment’ levels (bagged plants, ant-excluded plants, and plants exposed to pollinators and ants) and ‘species’ (*Dyckia floribunda* and *D. longipetala*) for both response variables: ‘fruit set’ and ‘seeds per plant’. Differences between the means (Estimate), standard deviations (SEs) of these differences, and *p*-values adjusted using the Tukey method are presented.

Variable	Contrast	Estimate	SE	*p*-Value
Fruit set	Bagged *D. floribunda*–Ant-excluded *D. floribunda*)	−0.253	0.051	8.63 × 10^−1^
	Bagged *D. floribunda*–Exposed *D. floribunda*	−0.452	0.051	4.29 × 10^−7^
	(Ant-excluded *D. floribunda*)–Exposed *D. floribunda*	−0.199	0.051	0.003
	Bagged *D. longipetala*–(Ant-excluded *D. longipetala*)	−0.077	0.050	0.634
	Bagged *D. longipetala*–Exposed *D. longipetala*	−0.252	0.051	9.06 × 10^−1^
	(Ant-excluded *D. longipetala*)–Exposed *D. longipetala*	−0.175	0.050	0.010
Seeds per plant	Bagged *D. floribunda*–(Ant-excluded *D. floribunda*)	−6965.26	2366.17	0.049
	Bagged *D. floribunda*–Exposed *D. floribunda*	−16,326.86	2366.17	5.14 × 10^−4^
	(Ant-excluded *D. floribunda*)–Exposed *D. floribunda*	−9361.59	2366.17	0.003
	Bagged *D. longipetala*–(Ant-excluded *D. longipetala*)	−3209.11	2301.94	0.730
	Bagged *D. longipetala*–Exposed *D. longipetala*	−11,851.02	2366.17	7.11 × 10^−1^
	(Ant-excluded *D. longipetala*)–Exposed *D. longipetala*	−8641.91	2301.94	0.005

## Data Availability

Data are contained within the article and Appendix A.

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
