# Peer review of "Plant Reproductive Success Mediated by Nectar Offered to Pollinators and Defensive Ants in Terrestrial Bromeliaceae"

_plants, 2024, doi:10.3390/plants13040493_

Round 1

Reviewer 1 Report

Comments and Suggestions for Authors

This manuscript describes experimental field work on two bromeliad species, testing the effects of exposure to natural pollinators with and without ant access to extrafloral nectaries on the flower stalk, relative to bagged stalks. The study system is appropriate and the results are interesting, though not well explained.

My chief concerns are:

- The hypothesis is not well justified in the introduction. There is a lot of published work that shows actual or potential interference between ants and pollinators, when EFNs occur on or near flowers. Adaptations for avoiding this conflict are expected and are also documented in the literature. However, the hypothesis presented in the manuscript goes beyond avoidance of conflict to predict that plants benefit from both pollinators and ants, even when they co-occur. The hypothesis should be better justified in the introduction, in the context of the literature.

- The findings are not explained mechanistically. The results demonstrate that the presence of ants enhances fruit and seed production in the presence of pollinators, but there is no attempt at a causal explanation of this result, which is surprising given that this result was predicted. Was the effect caused by ant-related change in pollination or in the fate of fruits and seeds post-pollination (or both)? The abstract mentions measuring pollinator visitation frequency; are there data that can be brought to bear on the question of how ant exposure affected pollinator visitation? Similarly, the fruits and seeds were harvested and counted, providing an opportunity to assess fruit and seed damage. There is a mention of fruit and seed damage in the discussion (lines 330-332). Was there an effect of ants on damage to these reproductive structures?

- The manuscript concludes with a paragraph on continuous nectar production; a topic which is not the focus of the study. The manuscript would benefit from a short concluding paragraph that summarizes what is novel and important about the findings in the context of what we know about similar systems.

More focused comments:

Lines 13-44 - Sentences starting on lines 24 and 31 of the abstract should be clarified. The first sentence of the introduction is not clear.

Lines 95-97 - The information presented here is puzzling. If this is a statement of fact, a citation is needed. And if the patterns of self-compatibility are known, it would be helpful to know why compatibility was tested in this study.

Line 101 - I don't think it is accurate, nor necessary, to call these model species.

Lines 116-117 - The statement about the cost of nectar secretion is speculative. It appears again in the discussion, which is a better place for it; I suggest deleting it from the methods.

Figure 1 - The photos in Fig. 1 are indistinct and not very helpful for the reader. In contrast, the photos in Figure 2 are quite informative. I suggest cutting Fig. 1.

Lines 131-137 - The section 2.2: Pollinators, ants and insects consuming plant tissues is premature here and should be moved after the basics of the sampling design have been introduced. Regarding the data collection described in the section, the reader needs to know where and when the observations were made, times of day or night, and the number of plants observed. How was deterrence of pollinators (which is mentioned in the results) ascertained?

Lines 138-154 - In section 2.3, the number of plants included in this manipulation should be clarified.

Line 157 - Here it says there were three populations studied, but Figure 3 shows data for five populations.

Lines 242-243 - Consider presenting data on pollen tube lengths to support the contention that no difference exists between cross- and self-pollinated. This could appear in the supplementary materials.

Line 245 - 246 - Report the model results in a table so the reader can assess the details, e.g. test statistic, degrees of freedom. Also, report actual p-values.

Table 1 - There appears to be some misreporting here. For both plant species, the values for the bagged treatment fruit set in Table 1 appear much too high - are these actually the values for seed set? The value reported for seed set for bagged longipetala appears very low - is this actually the data for fruit set? There is no data at all reported for the bagged treatment for floribunda. (Note also there are misspellings in the table.)

Lines 317-321 - If this is a point worth making, its relevance to the current study should be made clearer.

Lines 333 - 335 - The initial clause of this sentence is overly speculative in context.

Lines 339-341 - Similarly, the contention that nectar section is continuous should be properly qualified if it has not been measured (e.g. extranuptial nectar <add "appears to be"> secreted continuously in... since...

Lines 350-353 - The final sentence of the paper is overly vague. What concrete conclusion should we draw from the results of this study?

Comments on the Quality of English Language

The writing should be clarified, especially in the introduction and discussion. In some cases a lack of clarity is due to sentence construction, and in other cases the point is vague. There are numerous minor grammatical problems throughout the manuscript that should be corrected.

Author Response

Responses to Reviewer 1

This manuscript describes experimental field work on two bromeliad species, testing the effects of exposure to natural pollinators with and without ant access to extrafloral nectaries on the flower stalk, relative to bagged stalks. The study system is appropriate, and the results are interesting, though not well explained.

R: Thank for this general overview of our Ms and all the constructive comments and suggestions.

My chief concerns are:

- The hypothesis is not well justified in the introduction. There is a lot of published work that shows actual or potential interference between ants and pollinators, when EFNs occur on or near flowers. Adaptations for avoiding this conflict are expected and are also documented in the literature. However, the hypothesis presented in the manuscript goes beyond avoidance of conflict to predict that plants benefit from both pollinators and ants, even when they co-occur. The hypothesis should be better justified in the introduction, in the context of the literature.

R: Agreed. The hypothesis and expected results were improved according to this suggestion. In this study we analyzed the general effect of excluding pollinators and/or ants on the plant reproductive success. We acknowledge that animal-animal interactions (pollinators with ants) could also affect plant reproductive success (when ants deter pollinators, for instance), as well as other factors not considered in our experimental design. We have carefully considered this point to avoid any misunderstanding between our experimental design and the results presented throughout the Ms. We have also introduced this consideration at the end of the discussion for future studies with a mechanistic approach that could analyze the eventual interference between ants and pollinators generating a trade-off for plant reproductive success.

- The findings are not explained mechanistically. The results demonstrate that the presence of ants enhances fruit and seed production in the presence of pollinators, but there is no attempt at a causal explanation of this result, which is surprising given that this result was predicted. Was the effect caused by ant-related change in pollination or in the fate of fruits and seeds post-pollination (or both)? The abstract mentions measuring pollinator visitation frequency; are there data that can be brought to bear on the question of how ant exposure affected pollinator visitation? Similarly, the fruits and seeds were harvested and counted, providing an opportunity to assess fruit and seed damage. There is a mention of fruit and seed damage in the discussion (lines 330-332). Was there an effect of ants on damage to these reproductive structures?

R: Agreed. In this study we analyzed the general effect of excluding pollinators and/or ants on the plant reproductive success. We acknowledge that animal interactions (pollinators with ants) could also affect plant reproductive success (when ants deter pollinators, for instance), as well as other factors not considered in our experimental design. We have carefully considered this point to avoid any misunderstanding between our experimental design (hypothesis and expected results) and the results presented throughout the Ms. We have also introduced this consideration at the end of the discussion for future studies with a mechanistic approach that could analyze the eventual interference between ants and pollinators generating a trade-off for plant reproductive success. Frequency of pollinator visits was recorded for exposed plants (and not in comparison with ant-excluded plants). We agree that this comparison could be very interesting (particularly for small pollinators) but we need to perform observations separated several meters from the spikes because the main pollinators are hummingbirds. In consequence, detailed observation for ant-deterrence on small pollinators was not possible. Nevertheless, we acknowledge that this point in the discussion section. Finally, we measured the general decrease of fruit and seed sets. We have not detailed data on per fruit (and seed) damage because we harvested the spikes at the end of the season when all attacked flowers have been lost as well as many predated fruits at early developmental stages. Thus, we occasionally see larvae on flowers and fruits (and then reported that) but we do not perform a detailed protocol to quantify them because we want to know (first) if there is an ant-effect on reproductive success (i.e., an active defense of reproductive structures corroborated indirectly by final fruit- and seed-set).

- The manuscript concludes with a paragraph on continuous nectar production; a topic which is not the focus of the study. The manuscript would benefit from a short concluding paragraph that summarizes what is novel and important about the findings in the context of what we know about similar systems.

R: Agreed. Done. We have replaced the final sentence for a new concluding paragraph considering these suggestions.

More focused comments:

Lines 13-44 - Sentences starting on lines 24 and 31 of the abstract should be clarified. The first sentence of the introduction is not clear.

R: Agreed. Done. We have modified all these initial sentences of the abstract (highlighted in yellow). The abstract was shortened.

Lines 95-97 - The information presented here is puzzling. If this is a statement of fact, a citation is needed. And if the patterns of self-compatibility are known, it would be helpful to know why compatibility was tested in this study.

R: Agreed. Done. This information was clarified in both the Introduction and in the M&M sections.

Line 101 - I don't think it is accurate, nor necessary, to call these model species.

R: Agreed. Done. “model” was deleted.

Lines 116-117 - The statement about the cost of nectar secretion is speculative. It appears again in the discussion, which is a better place for it; I suggest deleting it from the methods.

R: Agreed. Done. The cost of nectar secretion in now presented only in the Discussion section.

Figure 1 - The photos in Fig. 1 are indistinct and not very helpful for the reader. In contrast, the photos in Figure 2 are quite informative. I suggest cutting Fig. 1.

R: Agreed. Done. The Fig. 1 was moved to supplementary material (now as SFig. 1). Nevertheless, if the Editor thinks it would be better to delete SFig. 1 from this section we also agree.

Lines 131-137 - The section 2.2: Pollinators, ants and insects consuming plant tissues is premature here and should be moved after the basics of the sampling design have been introduced. Regarding the data collection described in the section, the reader needs to know where and when the observations were made, times of day or night, and the number of plants observed. How was deterrence of pollinators (which is mentioned in the results) ascertained?

R: Agreed. Done. This section was moved and the information on data collection was added (the reviewer 2 also asked for this information).

Lines 138-154 - In section 2.3, the number of plants included in this manipulation should be clarified.

R: Agreed. Done. This information was added (the reviewer 2 also asked for this information).

Line 157 - Here it says there were three populations studied, but Figure 3 shows data for five populations.

R: The population are indicated with different letters (A to F) and colors in the Figure 3 (now new Figure 2). Thus, there are 6 population presented (3 for each species). A, B and C for one species and the other 3 letters for the other species. We have improved the legend for this figure.

Lines 242-243 - Consider presenting data on pollen tube lengths to support the contention that no difference exists between cross- and self-pollinated. This could appear in the supplementary materials.

R: We have not reported the pollen tube length in this case because all the tubes reach the ovules (i.e., the same length). It would be interesting to compare the growing rate of these two kinds of pollen tubes, but unfortunately these data are not available to be reported.

Line 245 - 246 - Report the model results in a table so the reader can assess the details, e.g. test statistic, degrees of freedom. Also, report actual p-values.

R: Agreed. Done. This information was originally included in STable 1. We have moved this Table to the Ms (now as Table 2) to present the information easily available for the reader.

Table 1 - There appears to be some misreporting here. For both plant species, the values for the bagged treatment fruit set in Table 1 appear much too high - are these actually the values for seed set? The value reported for seed set for bagged longipetala appears very low - is this actually the data for fruit set? There is no data at all reported for the bagged treatment for floribunda. (Note also there are misspellings in the table.)

R: Agreed. We have included Table 2 (in the previous version as a STable 1) to complete the information in the MS. We have added a line in this Table 1 to separate results between species to improve the presentation of the results for the reader. Values for the bagged treatment for fruit- and seed-set in D. longipetala are 0 (see Means column); the predicted values for the confidence interval are calculated by the statistical package (see M&M section). We have solved the misspellings in this Table (highlighted in yellow).

Lines 317-321 - If this is a point worth making, its relevance to the current study should be made clearer.

R: Agreed. Done. These sentences were deleted.

Lines 333 - 335 - The initial clause of this sentence is overly speculative in context.

R: Agreed. Done. These sentences were deleted.

Lines 339-341 - Similarly, the contention that nectar section is continuous should be properly qualified if it has not been measured (e.g. extranuptial nectar <add "appears to be"> secreted continuously in... since...

R: Agreed. Done. This sentence was modified according to this suggestion.

Lines 350-353 - The final sentence of the paper is overly vague. What concrete conclusion should we draw from the results of this study?

R: Agreed. Done. A conclusion was added and some of the suggestion for future studies mentioned by this reviewer was also included here.

The writing should be clarified, especially in the introduction and discussion. In some cases a lack of clarity is due to sentence construction, and in other cases the point is vague. There are numerous minor grammatical problems throughout the manuscript that should be corrected.

R: Agreed. The MS has been checked by a colleague with a fluent English and all the modifications has also been highlighted in yellow.

Reviewer 2 Report

Comments and Suggestions for Authors

The submitted work presents an important problem of interactions between flowering plants and animals. It describes the complex interactions between pollinators of two Dyckia species and ants, and their impact on the reproductive success of the studied populations. Additionally, the value of the work is increased by examining the impact of IC/SI on the background of the described interactions. It's a pity that this interesting aspect was not discussed in more detail in the Discussion.

My overall assessment of the work is positive, I would suggest making corrections to the M&M section. My few comments are marked directly on the attached PDF.

Author Response

Responses to Reviewer 2

The submitted work presents an important problem of interactions between flowering plants and animals. It describes the complex interactions between pollinators of two Dyckia species and ants, and their impact on the reproductive success of the studied populations. Additionally, the value of the work is increased by examining the impact of IC/SI on the background of the described interactions. It's a pity that this interesting aspect was not discussed in more detail in the Discussion.

R: Thank you for your positive comments. The discussion was improved considering these suggestions (and those from the Reviewer 1).

My overall assessment of the work is positive, I would suggest making corrections to the M&M section. My few comments are marked directly on the attached PDF.

R: Thank you for your detailed comments and suggestions that improved the new version of the MS. We have considered all of them in the new version (highlighted in yellow).
